# A cross-sectional investigation of SARS-CoV-2 seroprevalence and associated risk factors in children and adolescents in the United States

**Rebecca E. Levorson[1,2]\***, **Erica Christian[3]**, **Brett Hunter[4]**, **Jasdeep Sayal[3]**, **Jiayang Sun[4]**, **Scott A. Bruce[4]**, **Stephanie Garofalo[5]**, **Matthew Southerland[5]**, **Svetlana Ho[3]**, **Shira Levy[3]**, **Christopher Defillipi[5]**, **Lilian Peake[6]**, **Frederick C. Place[7ʘ]**, **Suchitra K. Hourigan**[3,8ʘ]

1 Division of Pediatric Infectious Diseases, Inova Children's Hospital, Falls Church, Virginia, United States of America, 2 Division of Pediatric Infectious Diseases, Pediatric Specialists of Virginia, Fairfax, Virginia, United States of America, 3 Division of Pediatric Research, Inova Children's Hospital, Falls Church, Virginia, United States of America, 4 Department of Statistics, George Mason University, Fairfax, Virginia, United States of America, 5 Inova Heart and Vascular Institute, Inova Health System, Falls Church, Virginia, United States of America, 6 Division of Epidemiology, Virginia Department of Health, Richmond, Virginia, United States of America, 7 Division of Pediatric Emergency Medicine, Inova Children's Hospital, Falls Church, Virginia, United States of America, 8 Laboratory of Host Immunity and Microbiome, National Institute of Allergy and Infectious Diseases, Bethesda, Maryland, United States of America

ʘ These authors contributed equally to this work.
* RLevorson@psvcare.org

## Abstract

### Background

Pediatric SARS-CoV-2 data remain limited and seropositivity rates in children were reported as <1% early in the pandemic. Seroepidemiologic evaluation of SARS-CoV-2 in children in a major metropolitan region of the US was performed.

### Methods

Children and adolescents ≤19 years were enrolled in a cross-sectional, observational study of SARS-CoV-2 seroprevalence from July-October 2020 in Northern Virginia, US. Demographic, health, and COVID-19 exposure information was collected, and blood analyzed for SARS-CoV-2 spike protein total antibody. Risk factors associated with SARS-CoV-2 seropositivity were analyzed. Orthogonal antibody testing was performed, and samples were evaluated for responses to different antigens.

### Results

In 1038 children, the anti-SARS-CoV-2 total antibody positivity rate was 8.5%. After multivariate logistic regression, significant risk factors included Hispanic ethnicity, public or absent insurance, a history of COVID-19 symptoms, exposure to person with COVID-19, a household member positive for SARS-CoV-2 and multi-family or apartment dwelling without a private entrance. 66% of seropositive children had no symptoms of COVID-19. Secondary analysis included orthogonal antibody testing with assays for 1) a receptor binding domain

**Data Availability Statement:** Data is contained within the paper and Supporting Information files.

**Funding:** The Virginia Department of Health provided funding for this study. The Virginia

Department of Health, in conjunction with the authors of this manuscript, were involved in study design, the writing of the report and the decision to submit the paper for publication. The Virginia Department of Health were not involved in collection, analysis or interpretation of the data. Research reported in this publication was also supported in part by the National Institute of Child Health and Human Development under Award Number K23HD099240 (Hourigan) and by the Intramural Research Program of the National Institute of Allergy and Infectious Diseases (Hourigan), of the National Institutes of Health. Research reported in this publication was also supported in part via a subcontract from the parent award by the National Center for Advancing Translational Sciences of the National Institutes of Health under Award Number UL1TR003015 (Hunter, Sun, Bruce). The content is solely the responsibility of the authors and does not necessarily represent the official views of the National Institutes of Health. The funders had no role in study design, data collection and analysis, decision to publish, or preparation of the manuscript.

**Competing interests:** Author Christopher Defillipi consults for Ortho Diagnostics, Abbott Diagnostics, and Siemens Healthineers. No other authors have any conflicts of interest relevant to this article to disclose.

specific antigen and 2) a nucleocapsid specific antigen had concordance rates of 80.5% and 79.3% respectively.

## Conclusions

A much higher burden of SARS-CoV-2 infection, as determined by seropositivity, was found in children than previously reported; this was also higher compared to adults in the same region at a similar time. Contrary to prior reports, we determined children shoulder a significant burden of COVID-19 infection. The role of children's disease transmission must be considered in COVID-19 mitigation strategies including vaccination.

## Introduction

Sustained community transmission of SARS-CoV-2 is indisputable in the United States (US) and globally. The extent of SARS-CoV-2 community transmission has remained incompletely delineated, particularly in children. Initially, our understanding was dependent upon aggregate molecular testing for acute infection and largely missed mild and asymptomatic infections. Based on this data, early reports misleadingly suggested that children were largely spared infection from SARS-CoV-2 [1, 2].

Seroprevalence estimates are useful to understand the cumulative incidence in a given population and provide data about asymptomatic or subclinical infections that would not otherwise be detected. Limited pediatric seroprevalence data exist for SARS-CoV-2. Following the first pandemic wave in Spain, up to a 6.2% seropositivity was reported late April/early May 2020, with only 3.8% of children being seropositive [3]. Antibody positivity in a smaller study in Switzerland in mid-May 2020 was lowest in children 0–9 years at 0.8%, compared with adults aged 20–49, at 9.9% [4]. Similarly, in Germany a low seropositivity rate of 0.6% was reported in children aged 1–10 years [5].

An investigation in spring 2020 evaluated children for SARS-CoV-2 antibodies 8–10 weeks after a large school outbreak in Chile. Anti-SARS-CoV-2 antibodies were present in 10% of children compared with 17% of adult staff. Forty percent of seropositive children were categorized as asymptomatic [6].

Since the conclusion of our study, several other pediatric US seroprevalences have been reported from residual blood in children. One reported a seroprevalence range of 2.5% to 16.3% between May and September 2020 [7]. Another in metro DC between July and October 2020 showed a seroprevalence of 9.5% in children undergoing venipuncture for another reason [8]. Limitations exist in both reports as to applicability to the entire pediatric population due to use of residual blood samples in children with other medical concerns.

Compared to adults, preliminary evidence in children suggested a distinct antibody response to SARS-CoV-2 infection with a reduced breadth of anti-SARS-CoV-2 antibodies, specifically a reduction in anti-nucleocapsid antibody production [9]. Furthermore, it has been postulated that children may have a higher prevalence of antibodies against the S2 unit of the SARS-CoV-2 spike protein, partially due to cross-reactivity to conserved epitopes found in benign seasonal human coronaviruses [10].

Studies to date have attempted to assess seroepidemiology of SARS-CoV-2 disease on select pediatric populations, however, broad sampling of the representative pediatric population is needed for more accurate assessment of true seroprevalence of disease. In such, the primary objective of this study was to determine SARS-CoV-2 seroprevalence after the first large wave

of disease in our community in children residing in Northern Virginia, US, a populous and urbanized region of Virginia. Secondary objectives included identification of risk factors associated with SARS-CoV-2 seropositivity in children, orthogonal antibody testing to assess the accuracy of antibody testing in children, and evaluation of immunologic responses to different antigens of SARS-CoV-2 in children.

## Methods

Children and adolescents were enrolled in this prospective cross-sectional, observational study, from July 31 to October 13, 2020. The Inova Health Systems Human Research Protection Office waived approval as per CFR 45 46.102, as this project was a public health surveillance activity. Verbal informed consent was obtained from parents (for children <18 years) and adolescents ≥18 years and documented by study staff in an electronic database. Assent was additionally obtained for those over 7 years of age. Written consent was not obtained as verbal consent was more feasible and this study was not deemed as research. During the study period, public schools in Northern Virginia were closed to in-person learning although some childcare facilities remained open. An indoor mask mandate outside the home was in effect but only for children ≥10 years and adults.

Subjects were recruited from three settings in order to allow for more representativeness of the entire pediatric population: 1) non-emergency health care settings (including pediatric primary care offices, pediatric specialty clinics and pre-surgical areas for elective procedures); 2) self-referral in response to advertisement of the study, and 3) the Inova Children's Hospital Pediatric Emergency Department and limited enrollment from inpatient units. Inclusion criteria were ≤19 years and residence in Virginia. Only one individual per household could enroll. Exclusion criteria included receipt of immunoglobulin therapy within the past 11 months.

Enrolled participants completed a questionnaire collecting demographic, health and potential COVID-19 exposure information including symptoms possibly consistent with COVID-19 infection (fever, cough, difficulty breathing, chills, muscle pain, sore throat, new onset diarrhea, or loss of taste or smell). Blood was collected by venipuncture, either for the sole purpose of the study or in conjunction with another blood draw. Blood was tested using the US Food and Drug Administration emergency use authorized Ortho Clinical Diagnostics VITROS Immunodiagnostic Products Anti-SARS-CoV-2 Total test (Ortho Clinical) performed on the Vitros 3600 system (Ortho Clinical Diagnostics, Raritan, NJ) to detect total antibody (IgG, IgA and IgM) responses against the SARS-CoV-2 spike protein. This assay has a reported clinical specificity of 100% (95% CI: 99.1–100%) [11]. Samples were documented as reactive (≥1.00 S/Co) or non-reactive (<1.00 S/Co) for anti-SARS-CoV-2.

Orthogonal antibody testing was performed on reactive samples as the estimated prevalence prior to the study was 1%, based on limited data at the time [4]. Orthogonal serologic testing was recommended by the CDC for populations with a low pre-test probability, including low or unknown prevalence of disease [12]. In such conditions, an initial antibody assay is performed, and then on positive tests, a secondary assay with a different or more focused target is performed. In this study, the Ortho Clinical assay (targeting the total protein—S1 and S2 subunits) was used as the primary assay for analysis, followed by the Siemens SARS-CoV-2 IgG assay solely targeting the S1 unit receptor binding domain (RBD) (Siemens Healthineers, Erlangen, Germany) [13]. Additionally, to assess the antibody response in children to different SARS-CoV-2 antigens, all initially reactive samples were tested using the Abbott SARS-CoV-2 IgG Architect antibody assay (Abbott Laboratories, Chicago, IL) which targets the nucleocapsid antigen [14]. Furthermore, a random 10% sample of the initial non-reactive samples on the Ortho Clinical assay were analyzed on the Siemens (RBD) and Abbott (nucleocapsid) IgG assays.

Sample size justification was based on early estimates suggesting a SARS-CoV-2 seroprevalence in children of approximately 1% [4]. Thus, in order to estimate the seropositivity in demographic subgroups while considering the clustering effect due to testing at different enrollment sites, 80% Wilson score intervals were used to determine a total sample size of at least 1,000 children.

Statistical analysis methods included both comprehensive descriptive statistics and statistical modeling. Positive frequency counts were used to estimate both the overall and subgroup prevalence rates to delineate the effects of different factors. The corresponding Wilson score intervals provided the uncertainty quantification of the prevalence estimates. Graphical diagnostics and chi-squared tests of independence were used to determine whether selected covariates were correlated. Univariate and multivariate logistic regression analyses of antibody presence were used to explore potential effects from covariates. A final logistic regression model was chosen using a stepwise model selection. Odds ratios with corresponding confidence intervals were calculated for the coefficients in the model. The distribution of antibody titer levels across different factor levels was examined. Kruskal-Wallis tests were used to determine if these distributions differed across potential risk factors. Simple inter-rater reliability rates were calculated to assess orthogonal testing results. R version 4.0.3 was used to perform statistical analyses.

## Results

This study included 1038 children. Demographic and clinical data are shown in Table 1. All age groups between 0–19 years were well represented, and racial and ethnic diversity was reflective of the Northern Virginia populace.

Table 1 shows the prevalence rates for anti-SARS-CoV-2 total antibody (Ortho Clinical assay) across demographic and clinical covariates of interest. The overall positivity rate for anti-SARS-CoV-2 total antibody was 8.5% (88/1038). SARS-CoV-2 antibodies were found in 8.2% of White children, 5.2% of Black or African American children, 5.7% of Asian children, and 16.2% of children identified as other racial origin. When compared by age groups a bimodal distribution was noted with a seropositive rate of 13.7% in young children 0–5 years, 7.5% in school-age children 6–10 years, 5.1% in early adolescents 11–15 years and 10.8% in older adolescents 16–19 years (Fig 1). There was no significant difference in positivity rate by month in which the study was conducted.

Especially high prevalence rates were noted in children with Hispanic ethnicity (26.8%, 55/207), those with public insurance (Medicaid, 21.6% 27/125), those without insurance (55.0%, 11/20), children living in multi-family or apartment dwellings without a private entrance (28.7%, 33/115), and children recruited from safety net primary care clinic generally serving uninsured and underserved families (primary care location 2, 25%, 16/64) (S1 Fig).

Of children exposed to an individual with a known history of COVID-19, 33.0% (35/106) had antibodies. Living in the same household with a person who tested positive for SARS-CoV-2 increased the seroprevalence rate to 52.5% (31/59). Having a personal history of symptoms consistent with COVID-19 was associated with a seroprevalence of 11.8% (30/255). However, 65.9% (58/88) of children with positive antibody testing had no personal history of symptoms, and 54.6% (48/88) had no known exposure. Two children in the study were previously hospitalized due to acute symptomatic COVID-19 infection.

After multiple regression, significant factors associated with seropositivity were found to include ethnicity, age group, insurance status, symptoms consistent with COVID-19, having a household member test positive for COVID-19 and dwelling type (S1 Table).

A final multivariate logistic regression model was chosen using stepwise model selection. Remaining significant predictors included Hispanic ethnicity, lack of insurance or public

**Table 1. Demographic information, prevalence rates, and 95% Wilson score intervals.**

| Variable | Category | Count | Estimated Prevalence Rate | 95% Wilson Score Intervals | |
|---|---|---|---|---|---|
| | | | | Lower Bound | Upper Bound |
| Overall | | 1038 | 8.5% | 6.9% | 10.3% |
| Gender | Male | 499 | 8.4% | 6.3% | 11.2% |
| | Female | 536 | 8.6% | 6.5% | 11.3% |
| | Not Disclosed | 3 | 0%* | 0% | 56.1% |
| Age Group | 0–5 years | 182 | 13.7% | 9.5% | 19.5% |
| | 6–10 years | 241 | 7.5% | 4.8% | 11.5% |
| | 11–15 years | 374 | 5.1% | 3.3% | 7.8% |
| | 16–19 years | 241 | 10.8% | 7.5% | 15.3% |
| Race | White | 717 | 8.2% | 6.4% | 10.5% |
| | Black or African American | 97 | 5.2% | 2.2% | 11.5% |
| | Asian | 106 | 5.7% | 2.6% | 11.8% |
| | American Indian or Alaska Native | 0 | --- | --- | --- |
| | Native Hawaiian or Other Pacific Islander | 2 | 0%* | 0% | 65.8% |
| | Other | 111 | 16.2% | 10.5% | 24.2% |
| | Unknown/Declined | 5 | 0%* | 0% | 43.4% |
| Ethnicity | Not Hispanic or Latino | 829 | 4.0% | 2.8% | 5.5% |
| | Hispanic or Latino | 207 | 26.6% | 21.0% | 33.0% |
| | Unknown | 2 | 0%* | 0% | 65.8% |
| Insurance | Private | 834 | 5.3% | 4.0% | 7.0% |
| | Medicaid | 125 | 21.6% | 15.3% | 29.6% |
| | Medicare | 8 | 12.5%* | 2.2% | 47.1% |
| | None or uninsured | 20 | 55.0% | 34.2% | 74.2% |
| | Other | 3 | 33.3%* | 6.1% | 79.2% |
| | Don't know | 4 | 25.0%* | 4.6% | 69.9% |
| | Military | 44 | 6.8% | 2.3% | 18.2% |
| Existing Medical Comorbidity | No | 807 | 8.3% | 6.6% | 10.4% |
| | Yes | 225 | 8.9% | 5.8% | 13.3% |
| | Don't Know | 3 | 0%* | 0% | 56.1% |
| | Missing | 3 | 33.3% | --- | --- |
| Symptoms | No | 774 | 7.2% | 5.6% | 9.3% |
| | Yes | 255 | 11.8% | 0% | 65.8% |
| | Unknown | 6 | 33.3%* | 9.7% | 70.0% |
| | Missing | 3 | 0% | --- | --- |
| Dwelling | Single-family | 901 | 5.9% | 4.5% | 7.6% |
| | Multi-family or apartment–No Private Entrance | 115 | 28.7% | 21.2% | 37.5% |
| | Multi-family or apartment–Private Entrance | 16 | 6.3% | 1.1% | 28.3% |
| | Other | 6 | 16.7%* | 3.0% | 56.4% |
| Household Member Exposure | Not Tested | 543 | 6.3% | 4.5% | 8.6% |
| | Tested–Positive | 59 | 52.5% | 40.0% | 64.7% |
| | Tested–Negative | 415 | 5.3% | 3.5% | 7.9% |
| | Tested–No Results | 15 | 6.7% | 1.2% | 29.8% |
| | Unknown if Tested | 4 | 0%* | 0% | 49.0% |
| | Missing | 2 | 0% | --- | --- |

(*Continued*)

**Table 1.** (Continued)

| Variable | Category | Count | Estimated Prevalence Rate | 95% Wilson Score Intervals | |
|---|---|---|---|---|---|
| | | | | Lower Bound | Upper Bound |
| COVID-19 Exposure | No | 911 | 5.6% | 4.3% | 7.3% |
| | Yes | 106 | 33.0% | 24.8% | 42.4% |
| | Unknown or Missing | 19 | 10.5% | 2.9% | 31.4% |
| | Missing | 2 | 0% | --- | --- |
| Reason for Visit | Specifically for COVID-19 antibody testing | 504 | 4.6% | 3.1% | 6.8% |
| | New or acute illness or injury | 193 | 14.0% | 9.8% | 19.6% |
| | Prevention or well visit | 59 | 20.3% | 12.0% | 32.3% |
| | Routine visit for a prior condition | 193 | 8.8% | 5.6% | 13.7% |
| | Other | 84 | 10.1% | 5.4% | 18.1% |
| Traveled Out of the State | No | 395 | 13.4% | 10.4% | 17.1% |
| | Yes | 642 | 5.5% | 3.9% | 7.5% |
| | Don't Know | 1 | 0%* | 0% | 79.3% |
| Traveled Out of the Country | No | 1002 | 8.4% | 6.8% | 10.3% |
| | Yes | 36 | 11.1% | 4.4% | 25.3% |
| Child Cared for outside of Home | No | 729 | 8.2% | 6.4% | 10.5% |
| | Yes | 299 | 8.7% | 6.0% | 12.4% |
| | Missing | 10 | 20.0% | --- | --- |
| Household Member Work Outside the Home | No work outside home | 383 | 6.3% | 4.2% | 9.2% |
| | Essential | 530 | 8.7% | 6.6% | 11.4% |
| | Unessential | 121 | 14.0% | 9.0% | 21.4% |
| | Unknown if essential | 3 | 33.3%* | 6.1% | 79.2% |
| | Missing | 1 | 0% | --- | --- |
| Enrollment Site | Pediatric Emergency Department | 175 | 14.3% | 9.9% | 20.2% |
| | Pre surgical areas | 89 | 6.7% | 3.1% | 13.9% |
| | Endoscopy suite | 10 | 0%* | 0% | 27.8% |
| | Pediatric inpatient units | 6 | 0%* | 0% | 39.0% |
| | Pediatric Intensive Care Unit | 7 | 14.3%* | 2.6% | 51.3% |
| | Pediatric Specialists Clinics | 444 | 4.7% | 3.1% | 7.1% |
| | Hematology/Oncology clinic | 57 | 17.5% | 9.8% | 29.4% |
| | Primary Care location 1 | 128 | 5.5% | 2.7% | 10.9% |
| | Primary Care location 2 | 64 | 25.0% | 16.0% | 36.8% |
| | Primary Care location 3 | 49 | 4.1% | 1.1% | 13.7% |
| | All other Primary Care clinics | 9 | 0%* | 0% | 29.9% |
| Enrollment Setting | Non-Emergency or non-inpatient setting | 346 | 11.3% | 8.4% | 15.0% |
| | Emergency or inpatient setting | 188 | 13.8% | 9.6% | 19.5% |
| | Self-referred | 504 | 4.6% | 3.1% | 6.8% |

*Indicates an unreliable estimate due to a small count.

insurance, a history of symptoms suggestive of COVID-19, exposure to an individual with a history of COVID-19, a household member testing positive for COVID-19, and living in a multi-family or apartment dwelling without private entrance (Table 2).

Of those who tested positive for anti-SARS-CoV-2, titer levels ranged from 1.05 to 1000 S/ Co. Across all age groups, titer levels did not significantly differ (Kruskal-Wallis, p = 0.088) (S2 Fig). There was no significant difference in titer levels between those with no symptoms, those

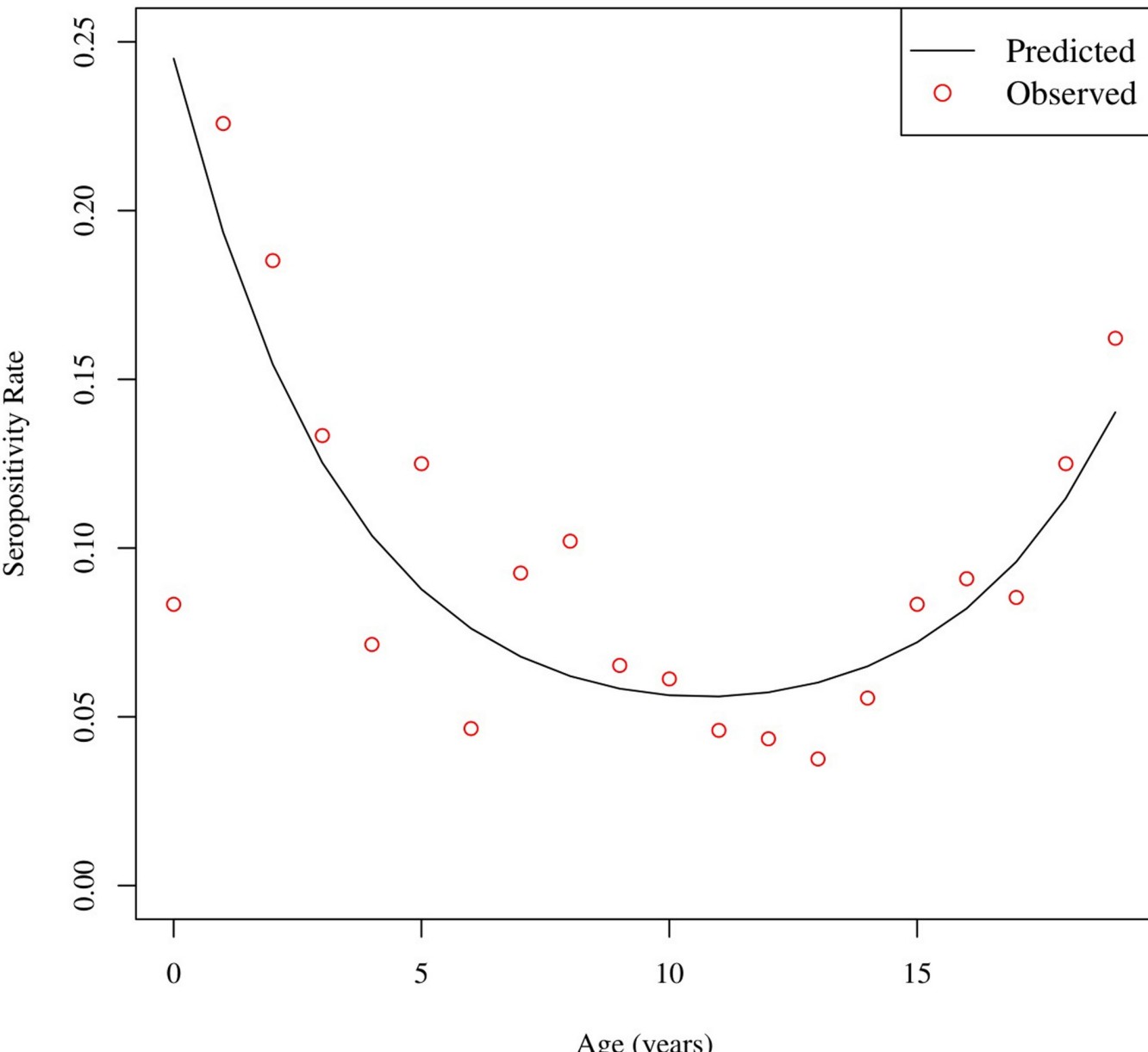

**Fig 1. Pairwise plot of age vs. seropositivity rate (%).** The red points represent the observed positivity rate in the cohort at each age, and the line is the predicted positivity rate produced by a quadratic regression model. The estimated seropositivity rate is higher for younger and older ages of children.

with symptoms within 2 weeks of testing, and those with symptoms more than 2 weeks before testing (p-value = 0.893, Kruskal-Wallis) (S3 Fig).

Orthogonal testing was performed on 87/88 reactive samples (where enough serum remained) using the Siemens assay against the S1/RBD target. This demonstrated an 80.5% agreement with 70/87 samples reactive. Those with lower titer levels on the Ortho Clinical assay were more likely to test negative on the Siemens assay (p = 0.001, logistic regression) and

**Table 2. Odds ratio estimates and p-values for chosen covariates.**

| Covariate | Level | Univariate Analysis | | Multivariate Analysis | |
|---|---|---|---|---|---|
| | | OR (95% CI) | p-value | OR (95% CI) | p-value |
| Ethnicity | Not Hispanic or Latino | Baseline | | Baseline | |
| | Hispanic or Latino | 8.73 (5.51, 14.02) | < 0.001 | 4.22 (2.26, 7.94) | < 0.001 |
| | Unknown | --- | 0.985 | --- | 0.994 |
| Insurance | Private | Baseline | | Baseline | |
| | Medicaid | 4.95 (2.91, 8.31) | < 0.001 | 2.45 (1.10, 5.39) | 0.027 |
| | Medicare | 2.56 (0.14, 14.86) | 0.383 | 0.47 (0.01, 7.05) | 0.628 |
| | None or uninsured | 21.94 (8.65, 57.16) | < 0.001 | 11.95 (3.71, 38.43) | < 0.001 |
| | Other | 8.98 (0.41, 95.46) | 0.075 | 5.11 (0.22, 60.29) | 0.206 |
| | Don't know | 5.98 (0.29, 47.83) | 0.125 | 1.93 (0.08, 19.29) | 0.603 |
| | Military | 1.31 (0.31, 3.80) | 0.659 | 3.37 (0.73, 11.40) | 0.075 |
| Symptoms | No | Baseline | | Baseline | |
| | Yes | 1.71 (1.06, 2.71) | 0.025 | 1.99 (1.05, 3.72) | 0.033 |
| | Unknown | 6.41 (0.88, 33.59) | 0.034 | 4.36 (0.27, 53.50) | 0.292 |
| Dwelling | Single-family | Baseline | | Baseline | |
| | Multi-family–no private entrance | 6.44 (3.92, 10.48) | < 0.001 | 3.19 (1.57, 6.43) | 0.001 |
| | Multi-family–private entrance | 1.07 (0.06, 5.42) | 0.951 | 0.71 (0.04, 4.63) | 0.765 |
| | Other | 3.20 (0.17, 20.31) | 0.292 | 1.15 (0.03, 17.95) | 0.931 |
| Household Member COVID-19 Test Results | Not Tested | Baseline | | Baseline | |
| | Tested–Positive | 16.57 (8.98, 31.02) | < 0.001 | 9.65 (3.34, 30.55) | < 0.001 |
| | Tested–Negative | 0.84 (0.48, 1.45) | 0.531 | 0.69 (0.35, 1.33) | 0.281 |
| | Tested–No Results | 1.07 (0.06, 5.57) | 0.949 | 2.36 (0.12, 15.00) | 0.446 |
| | Unknown if Tested | --- | 0.986 | --- | 0.993 |
| Child Exposed to COVID-19 | No | Baseline | | Baseline | |
| | Yes | 8.31 (5.06, 13.60) | < 0.001 | 2.93 (1.06, 7.45) | 0.030 |
| | Unknown | 1.98 (0.31, 7.18) | 0.368 | 3.22 (0.46, 13.79) | 0.160 |
| Travel Out of State | No | Baseline | | Baseline | |
| | Yes | 0.37 (0.24, 0.58) | < 0.001 | 0.53 (0.28, 0.98) | 0.045 |
| | Unknown | --- | 0.983 | --- | 0.995 |
| Number of Children in Household | | 0.84 (0.66, 1.06) | 0.154 | 0.67 (0.48, 0.92) | 0.016 |

older individuals were more likely to test negative (p = 0.011, logistic regression); Fig 2A. There was no significant difference between those with recent symptoms within 2 weeks versus those with more remote symptoms in negative results on orthogonal testing.

When comparing reactive samples from the Ortho Clinical assay with the Abbott assay targeting the nucleocapsid antigen there was a 79.3% agreement with 69/87 samples reactive. Those in older age groups were more likely to test negative on the Abbott assay (p = 0.001, logistic regression) but there was no significant change among titer levels (p = 0.177, logistic regression); Fig 2B.

A random selection of 98 samples (approximately 10%) that tested non-reactive on the Ortho Clinical assay were tested on both the Abbott and Siemens assays. All samples were negative on the additional two assays.

## Discussion

Using the Ortho Clinical anti-SARS-CoV-2 antibody assay, it was determined that the seroprevalence of SARS-CoV-2 in children in Northern Virginia was as high as 8.5% (95% CI 6.9–

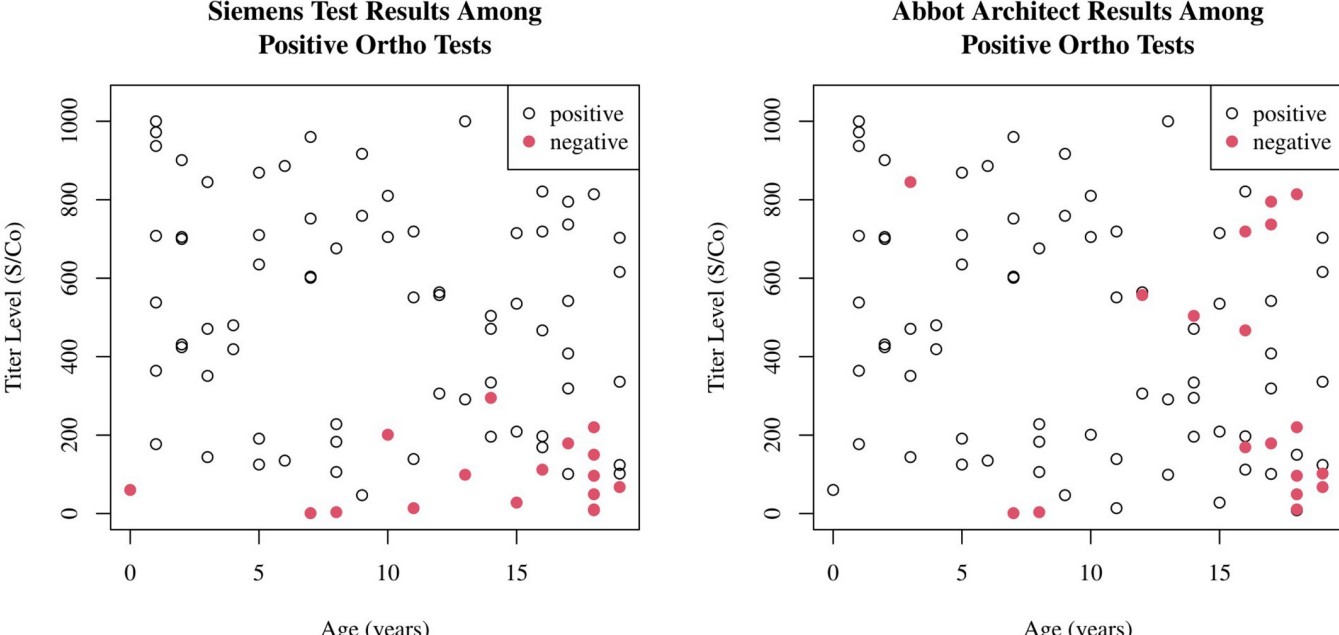

**Fig 2. Scatterplots showing age versus titer levels of the reactive tests on the Ortho Clinical Diagnostic Vitros assay, with orthogonal testing on the Siemens assay (2a) and Abbot assay (2b).** Negative tests on orthogonal testing are colored red.

10.3) in the fall of 2020. In contrast, the concurrently reported prevalence of SARS-CoV-2 in children in Virginia and throughout the US, based on molecular and antigen tests at the time of study completion, was around 1% [15].

An independently determined Northern Virginia adult seroprevalence was 4.4% in a similar study that concluded 2 months prior to our study [16]. As of mid-October, the CDC reported a Virginia adult seroprevalence of 4.1% [17]. The seropositivity rate in children in our study was more than double the reported concurrent adult seroprevalence. However, direct comparison between these rates is difficult because different study recruitment methodologies and antibody assays were used. The higher-than-expected pediatric seropositivity rate in our study is likely partly attributable to continued viral transmission over time. In addition, early in the pandemic, it was perceived that children were at reduced risk for acquiring COVID-19. However, the unrecognized burden of mild or asymptomatic disease, resulted in testing bias and under-representation of children in testing schemes [1, 18–20]. The high SARS-CoV-2 seropositivity in children identified was unexpected and has important implications for this silent burden of disease and risk of transmission to and from children.

Children under 5 were more likely to be seropositive for SARS-CoV-2 compared with other age groups in univariate analysis. This finding was surprising and has implications for the need for different considerations in mitigation and protection strategies in the younger age groups. Young children are cared for more closely by adults with potential higher risk of exposure and possibly interact with each other with less protection. Additionally, even though there was an indoor mask mandate outside the home, this was only for children ≥10 years. Moreover, even as public schools were closed for in-person learning, some childcare facilities remained open. However, consistent with other reports, our analyses did not indicate a significant difference in seropositivity in children cared for outside of their home, possibly as efforts are taken to prevent transmission in childcare settings [21].

The majority of children with anti-SARS-CoV-2 antibodies in this cohort (66%) did not recall having symptoms of COVID-19. While this can be affected by recall bias, this highlights the high proportion of infections that are asymptomatic in children.

For seropositive individuals, mean antibody titers did not appear to differ between those with or without symptoms, recent or remote. It is interesting that for some seropositive children, symptoms were reported <2 weeks from testing as the reported kinetics of the antibody response to SARS-CoV-2 demonstrate that at least a week is necessary to mount IgM responses after onset of symptomatic disease [20]. Since COVID-19 symptoms overlap greatly with other childhood viruses, it is possible that these reported symptoms were not truly caused by COVID-19 infection. It is also possible that some children with remote symptomatology had seroreversion and so were not captured [22, 23].

Exposure to infectious persons with COVID-19 is a well-recognized risk factor for acquiring COVID-19 as we confirmed [7, 24–26]. As previously reported, if the infectious person is a member of the same household the risk is even higher; over 50% of children exposed to a test-positive household member were seropositive [27]. Conversely, almost 60% of seropositive children had no known contact with test-positive SARS-CoV-2. This emphasizes the insidious effects of asymptomatic spread with viral shedding in symptomatic and asymptomatic individuals [28].

Ethnically and socioeconomically marginalized groups unable to shelter in place have shouldered a disproportionate burden of COVID-19 disease globally [27, 29, 30]. Hispanic ethnicity conferred four-fold odds of seropositivity. Publicly provided or lack of health insurance is an accepted marker for lower socioeconomic status, and families with limited financial means are more likely to live in multi-family dwellings. Significantly higher seropositivity rates in each of these groups demonstrated the disproportionate impact of COVID-19 on the underprivileged.

Orthogonal testing with an antibody assay specific to the S1/RBD antigen of the spike protein demonstrated a concordance of 80.5%. Those samples which failed to show concordance in orthogonal testing to the S1/RBD exhibited very low titers to the full spike protein (Ortho Clinical assay). Future consideration should be given to redefining the lower limit of detection for full spike protein antibody assays as it is possible that these lower titer levels may represent cross-reactivity to conserved epitopes in benign seasonal human coronaviruses [10, 31], with the Ortho Clinical assay overestimating the seroprevalence in children; however it is not possible to confirm this in our study cohort. Another possibility for discordant results may be there is reduced SARS-CoV-2 neutralizing activity among children given that most neutralizing epitope targets are found in the S1/RBD portion of S protein. In addition, some samples that may have been negative on orthogonal testing may have been due to the Ortho Clinical assay also detecting IgM; however those negative on orthogonal testing were not more likely to have recent symptoms consistent with COVID-19.

Children with a response to the full spike protein assay also frequently manifested antibodies to the nucleocapsid antigen (79.3% concordance). A possible reason why a higher concordance was not found is that children may have a decreased antibody response to the nucleocapsid antigen, compared to adults, as previously reported [9]. Proposed biological reasons for this include a lower release of nucleocapsid proteins related to lower replication in children [9], different viral pathogenesis in children compared with adults and possibly a need for a lower sensitivity of serological detection when using nucleocapsid based assays alone in children [32]. Interestingly, it was those in older age groups of children that more likely failed to produce a nucleocapsid antibody response, raising the possibility that the older age group may be more likely to have a false positive or IgM positive response on the Ortho Clinical assay.

Given all the tested negative samples on the Ortho Clinical assay were also negative on both the assay specific to the S1/RBD antigen of the spike protein and the nucleocapsid antigen, suggests a high specificity of these subsequent assays.

## Limitations

There are several potential limitations to this investigation. Selection bias may have affected the representativeness of the regional population, as it was focused on self-referral and children having blood drawn for another clinical purpose. However, these factors are accounted for in this analysis. In addition, Northern Virginia is a major metropolitan area, and so results may not be generalizable to more rural areas. As noted, cross-reaction with other common endemic human beta-coronaviruses may result in false positive antibody results, particularly at currently defined thresholds [10]. Conversely, seroreversion may produce false negative results [27, 28]. Most importantly, this study represents a static snapshot of a dynamic period [4]. In addition, true prevalence rates using the Siemens (RBD) and Abbot (nucleocapsid) IgG assays individually cannot be calculated as not all study samples were run on all assays.

## Conclusions

Over six months into this pandemic, a much higher pediatric SARS-CoV-2 burden was found than had been reported by viral testing in the US by the end of 2020; this rate was also higher than in adults in the same region over a similar time period. These seropositive rates followed the period after the first peak of infection in the community, when schools were closed and social mitigation strategies were mandated. Contrary to early reporting in the COVID-19 pandemic, this study determined that children experience a high rate of COVID-19 infection.

This study also characterized and reinforced epidemiologic data demonstrating the inequitable burden of COVID-19 on disadvantaged socioeconomic groups. Independent risk factors for disease included Hispanic ethnicity, lack of insurance or public insurance, and living in a multi-family or apartment dwelling without private entrance. Additionally, a history of symptoms suggestive of COVID-19, exposure to an individual with a history of COVID-19, a household member testing positive for COVID-19 were also significant risk factors for acquiring disease. Therefore, protection of children through continued social mitigation strategies and extention of COVID-19 vaccines to children are imperative to achieving control of this pandemic.

## Supporting information

**S1 Fig. Wilson score intervals for selected covariates of interest.**
(TIFF)

**S2 Fig. A multiple boxplot that shows the SARS-CoV-2 total spike protein titer levels (S/Co) for those who tested positive separated by age group and summary statistics.**
(TIFF)

**S3 Fig. A multiple boxplot that showing SARS-CoV-2 total spike protein titer levels (S/Co) for those who tested positive separated by presence of symptoms and summary statistics.**
(TIFF)

**S1 Table. Univariate and multivariate logistic regression results.**
(DOCX)

## Acknowledgments

We recognize the efforts of all team members at the multiple study sites who supported this study and encouraged participation. This project would not have been possible without the support of Dr. David Trump and the Virginia Department of Health and the pediatric teams at Farrell Pediatrics, Inova Cares Clinic for Children, Fairfax Pediatrics, Advanced Pediatrics, Pediatric Specialists of Virginia, and Inova Children's Hospital. We recognize and thank all of the participants of the study and their families; such studies can only happen because people find that helping in research, especially in the midst of a pandemic, is worthwhile, and therefore, choose to join.

## Author Contributions

**Conceptualization:** Rebecca E. Levorson, Christopher Defillipi, Lilian Peake, Frederick C. Place, Suchitra K. Hourigan.

**Data curation:** Rebecca E. Levorson, Erica Christian, Jasdeep Sayal, Stephanie Garofalo, Svetlana Ho, Shira Levy, Frederick C. Place, Suchitra K. Hourigan.

**Formal analysis:** Brett Hunter, Jiayang Sun, Scott A. Bruce.

**Investigation:** Stephanie Garofalo, Matthew Southerland.

**Project administration:** Jasdeep Sayal, Stephanie Garofalo.

**Supervision:** Rebecca E. Levorson.

**Writing – original draft:** Rebecca E. Levorson, Brett Hunter, Frederick C. Place, Suchitra K. Hourigan.

**Writing – review & editing:** Rebecca E. Levorson, Erica Christian, Brett Hunter, Jasdeep Sayal, Jiayang Sun, Scott A. Bruce, Stephanie Garofalo, Matthew Southerland, Svetlana Ho, Shira Levy, Christopher Defillipi, Lilian Peake, Frederick C. Place, Suchitra K. Hourigan.

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
