## [Decision Letter · Decision Letter 0]

5 Oct 2021

PONE-D-21-25774SARS-CoV-2 Seroepidemiology in ChildrenPLOS ONE

Dear Dr. Hourigan,

Thank you for submitting your manuscript to PLOS ONE. After careful consideration, we feel that it has merit but does not fully meet PLOS ONE’s publication criteria as it currently stands. Therefore, we invite you to submit a revised version of the manuscript that addresses the points raised during the review process. Both reviewers has very minor comments. Please address them and submit ASAP. 

We look forward to receiving your revised manuscript.

Kind regards,

Gheyath K. Nasrallah

Academic Editor

PLOS ONE

2. Please modify the title to ensure that it is meeting PLOS’ guidelines (https://journals.plos.org/plosone/s/submission-guidelines#loc-title). In particular, the title should be "specific, descriptive, concise, and comprehensible to readers outside the field" and in this case  it does not meet our requirements in the current form and it is not informative and specific about your study's scope and methodology.

3. Please amend your current ethics statement to address the following concerns: Please explain why written consent was not obtained, how you recorded/documented participant consent, and if the ethics committees/IRBs approved this consent procedure.In addition, please confirm that consent was obtained from parents or guardians of children <18.

4. Thank you for stating the following in the Funding Section of your manuscript:

“The Virginia Department of Health provided funding for this study. The Virginia Department of Health, in conjunction with the authors of this manuscript, were involved in study design, the writing of the report and the decision to submit the paper for publication. The Virginia Department of Health were not involved in collection, analysis or interpretation of the data.  Research reported in this publication was also supported in part by the National Institute of Child Health and Human Development under Award Number K23HD099240 (Hourigan). Research reported in this publication was also supported in part via a subcontract from the parent award by the National Center for Advancing Translational Sciences of the National Institutes of Health under Award Number UL1TR003015 (Hunter, Sun, Bruce). The content is solely the responsibility of the authors and does not necessarily represent the official views of the National Institutes of Health”

We note that you have provided formation within the Funding. Please note that funding information should not appear in the other areas of your manuscript. We will only publish funding information present in the Funding Statement section of the online submission form.

“The Virginia Department of Health provided funding for this study. The Virginia Department of Health, in conjunction with the authors of this manuscript, were involved in study design, the writing of the report and the decision to submit the paper for publication. The Virginia Department of Health were not involved in collection, analysis or interpretation of the data.  Research reported in this publication was also supported in part by the National Institute of Child Health and Human Development under Award Number K23HD099240 (Hourigan). Research reported in this publication was also supported in part via a subcontract from the parent award by the National Center for Advancing Translational Sciences of the National Institutes of Health under Award Number UL1TR003015 (Hunter, Sun, Bruce). The content is solely the responsibility of the authors and does not necessarily represent the official views of the National Institutes of Health. The funders had no role in study design, data collection and analysis, decision to publish, or preparation of the manuscript.”

5. Please amend either the title on the online submission form (via Edit Submission) or the title in the manuscript so that they are identical.

6.Please review your reference list to ensure that it is complete and correct. If you have cited papers that have been retracted, please include the rationale for doing so in the manuscript text, or remove these references and replace them with relevant current references. Any changes to the reference list should be mentioned in the rebuttal letter that accompanies your revised manuscript. If you need to cite a retracted article, indicate the article’s retracted status in the References list and also include a citation and full reference for the retraction notice.

Reviewers' comments:

Reviewer's Responses to Questions

**Comments to the Author**

1. Is the manuscript technically sound, and do the data support the conclusions?

Reviewer #1: Yes

Reviewer #2: Yes

2. Has the statistical analysis been performed appropriately and rigorously? 

Reviewer #1: Yes

Reviewer #2: Yes

3. Have the authors made all data underlying the findings in their manuscript fully available?

Reviewer #1: Yes

Reviewer #2: Yes

4. Is the manuscript presented in an intelligible fashion and written in standard English?

Reviewer #1: Yes

Reviewer #2: Yes

5. Review Comments to the Author

Reviewer #1: The authors performed seroepidemiologic evaluation of SARS-CoV-2 in children in a major metropolitan region of the US. Three FDA approved serology tests were used to assess seroprevalence, Vitros as a primary assay, followed by Orthogonal testing with Siemens and Architect. The observed seroprevalence was higher than what is reported in the literature, I have questions in this regard:

1- Vitros (ortho clinical assay) measures the total antibody to the full spike protein, do you think that the use of Vitros as a primary assay for screening overestimated the seroprevalence in children? as you mentioned in the discussion line 291 “these lower titers levels may represent cross-reactivity to conserved epitopes in benign seasonal human coronavirus” which I agree with. I suggest you cite this paper to support your findings (JCI Insight. 2021;6(4):e144499. https://doi.org/10.1172/jci. insight.144499.).

Reviewer #2: The manuscript by Suchitra K Hourigan and colleagues, titled “SARS-CoV-2 Seroepidemiology in Children"presents a study and test performance data from SARS-CoV-2 spike protein levels in children and adolescents ≤19 years. The work was well-researched, and the authors mined good techniques to evaluate the performance of the assays in detecting SARS-CoV-2 antibodies. There are minor comment:

1#Since the samples were taken in the period July-Octobar 2020. It would be good to plot this months to see which month was most effected to link this date to occasion holidays or schools.

6. PLOS authors have the option to publish the peer review history of their article (what does this mean?). If published, this will include your full peer review and any attached files.

Reviewer #1: No

Reviewer #2: No

---

## [Author Response · Author response to Decision Letter 0]

18 Oct 2021

Response to Reviewers comments

Reviewer #1: The authors performed seroepidemiologic evaluation of SARS-CoV-2 in children in a major metropolitan region of the US. Three FDA approved serology tests were used to assess seroprevalence, Vitros as a primary assay, followed by Orthogonal testing with Siemens and Architect. The observed seroprevalence was higher than what is reported in the literature, I have questions in this regard:

1- Vitros (ortho clinical assay) measures the total antibody to the full spike protein, do you think that the use of Vitros as a primary assay for screening overestimated the seroprevalence in children? as you mentioned in the discussion line 291 “these lower titers levels may represent cross-reactivity to conserved epitopes in benign seasonal human coronavirus” which I agree with. I suggest you cite this paper to support your findings (JCI Insight. 2021;6(4):e144499. https://doi.org/10.1172/jci. insight.144499.).

Thank you for taking the time to review out manuscript. We are in agreement with your insightful comment. We have expanded line 291 to now read “ Future consideration should be given to redefining the lower limit of detection for full spike protein antibody assays as it is possible that these lower titer levels may represent cross-reactivity to conserved epitopes in benign seasonal human coronaviruses [10, 31], with the Ortho Clinical assay overestimating the seroprevalence in children; however it is not possible to confirm this in our study cohort.” and cited the additional paper you recommended (31).

Reviewer #2: The manuscript by Suchitra K Hourigan and colleagues, titled “SARS-CoV-2 Seroepidemiology in Children"presents a study and test performance data from SARS-CoV-2 spike protein levels in children and adolescents ≤19 years. The work was well-researched, and the authors mined good techniques to evaluate the performance of the assays in detecting SARS-CoV-2 antibodies. There are minor comment:

1#Since the samples were taken in the period July-Octobar 2020. It would be good to plot this months to see which month was most effected to link this date to occasion holidays or schools.

Thank you for taking the time to review out manuscript. This is an interesting point which we looked in to further. The positivity rate by month is as follows 

July: 0% (0/2)

August: 8.49% (46/542)

September: 10.03% (29/289)

October: 6.34% (13/205)

There was no statistically significant difference in rates by month and therefore we did not think it was worthwhile having a separate figure for this. However, we have added the following to the text, line 222 “There was no significant difference in positivity rate by month in which the study was conducted.”

---

## [Editor Report · Decision Letter 1]

27 Oct 2021

A cross-sectional investigation of SARS-CoV-2 seroprevalence and associated risk factors in children and adolescents in the United States

PONE-D-21-25774R1

Dear Dr. Suchitra K Hourigan

We’re pleased to inform you that your manuscript has been judged scientifically suitable for publication and will be formally accepted for publication once it meets all outstanding technical requirements.

Kind regards,

Gheyath K. Nasrallah

Academic Editor

PLOS ONE
---

## [Editor Report · Acceptance letter]

28 Oct 2021

PONE-D-21-25774R1 

A cross-sectional investigation of SARS-CoV-2 seroprevalence and associated risk factors in children and adolescents in the United States 

Dear Dr. Hourigan:

I'm pleased to inform you that your manuscript has been deemed suitable for publication in PLOS ONE. Congratulations! Your manuscript is now with our production department. 

Kind regards, 

on behalf of

Dr. Gheyath K. Nasrallah 

Academic Editor

PLOS ONE